# Why Did All Patients with Atrial Fibrillation and High Risk of Stroke Not Receive Oral Anticoagulants? Results of the Polish Atrial Fibrillation (POL-AF) Registry

**DOI:** 10.3390/jcm10194611

**Published:** 2021-10-08

**Authors:** Anna Szpotowicz, Iwona Gorczyca, Olga Jelonek, Beata Uziębło-Życzkowska, Małgorzata Maciorowska, Maciej Wójcik, Robert Błaszczyk, Agnieszka Kapłon-Cieślicka, Monika Gawałko, Monika Budnik, Tomasz Tokarek, Renata Rajtar-Salwa, Jacek Bil, Michał Wojewódzki, Janusz Bednarski, Elwira Bakuła-Ostalska, Anna Tomaszuk-Kazberuk, Anna Szyszkowska, Marcin Wełnicki, Artur Mamcarz, Małgorzata Krzciuk, Beata Wożakowska-Kapłon

**Affiliations:** 1Department of Cardiology, Regional Hospital, 27-400 Ostrowiec Swietokrzyski, Poland; szpotowiczanna@wp.pl (A.S.); mekrzciuk@gmail.com (M.K.); 21st Clinic of Cardiology and Electrotherapy, Swietokrzyskie Cardiology Centre, 25-736 Kielce, Poland; olga_jelonek@wp.pl (O.J.); bw.kaplon@poczta.onet.pl (B.W.-K.); 3Collegium Medicum, The Jan Kochanowski University, 25-369 Kielce, Poland; 4Department of Cardiology and Internal Diseases, Military Institute of Medicine, 04-141 Warsaw, Poland; buzieblo-zyczkowska@wim.mil.pl (B.U.-Ż.); mmaciorowska@wim.mil.pl (M.M.); 5Department of Cardiology, Medical University of Lublin, 20-059 Lublin, Poland; m.wojcik@umlub.pl (M.W.); robertblaszczyk1@wp.pl (R.B.); 61st Chair and Department of Cardiology, Medical University of Warsaw, 02-097 Warsaw, Poland; agnieszka.kaplon@gmail.com (A.K.-C.); mongawalko@gmail.com (M.G.); moni.budnik@gmail.com (M.B.); 7West German Heart and Vascular Centre, Institute of Pharmacology, University Duisburg-Essen, 45141 Essen, Germany; 8Department of Cardiology, Cardiovascular Research Institute Maastricht, Maastricht University Medical Centre, 6211LK Maastricht, The Netherlands; 9Department of Cardiology and Cardiovascular Interventions, University Hospital, 30-688 Krakow, Poland; tomek.tokarek@gmail.com (T.T.); rajfura@op.pl (R.R.-S.); 10Centre of Postgraduate Medical Education, Department of Invasive Cardiology, Central Clinical Hospital of the Ministry of Interior and Administration, 02-507 Warsaw, Poland; biljacek@gmail.com (J.B.); michaljerzywojewodzki@gmail.com (M.W.); 11Department of Cardiology, St John Paul II Western Hospital, Clinic of Cardiology, 05-825 Grodzisk Mazowiecki, Poland; medbed@wp.pl (J.B.); elwira.bakula@gmail.com (E.B.-O.); 12Medical Department, Lazarski University, 02-662 Warsaw, Poland; 13Department of Cardiology, Medical University, 15-276 Bialystok, Poland; a.tomaszuk@poczta.fm (A.T.-K.); annaszyszkowska92@gmail.com (A.S.); 143rd Department of Internal Diseases and Cardiology, Warsaw Medical University, 02-097 Warsaw, Poland; welnicki.marcin@gmail.com (M.W.); artur.mamcarz@wum.edu.pl (A.M.)

**Keywords:** atrial fibrillation, non-vitamin K antagonist oral anticoagulants, oral anticoagulants, stroke risk, vitamin K antagonists

## Abstract

Background: Most atrial fibrillation (AF) patients are at high risk of thromboembolic, and the use of oral anticoagulants (OACs) is advised in such cases. The aim of the study was to evaluate the frequency at which OACs were used in patients with AF and high risk thromboembolic complications, and identify factors that result in OACs not being used in the researched group of patients. Methods: The prospective, multicenter and non-interventional POL-AF registry is a study that includes AF patients from ten Polish cardiology centers. They were consecutively hospitalized between January and December of 2019. All the patients in the study were of high stroke risk. Results: A total of 3614 patients with AF and high stroke risk were included. Among the total study population, 91.5% received OAC therapy; antiplatelet therapy was prescribed for 3.7% of patients, heparin for 2.7%, and 2.1% of patients did not receive any stroke prevention therapy. Independent predictors of no OAC prescription were intracranial bleeding (OR 0.15, 95%CI 0.07–0.35, *p* < 0.001), gastrointestinal bleeding (OR 0.25, 95%CI 0.17–0.37, *p* < 0.001), cancer (OR 0.37, 95%CI 0.25–0.55, *p* < 0.001), hospitalization due to acute coronary syndrome (OR 0.48, 95%CI 0.33–0.69, *p* < 0.001), and anemia (OR 0.62, 95%CI 0.48–0.81, *p* < 0.001). Conclusions: Most AF patients with a high thromboembolic risk received OACs. The factors predisposing a lack of OAC use in these patients were conditions that significantly increased the risk of bleeding complications.

## 1. Introduction

Thromboembolic complications are the most serious implications of atrial fibrillation (AF) [1,2]. The danger of thromboembolic complications in patients with AF is not homogeneous and depends on age, sex, and concomitant diseases [3,4]. To evaluate the risk of thromboembolic complications, scores including risk factors of thromboembolic complications should be used [5]. The score recommended to evaluate the thromboembolic risk in patients with AF is the CHA_2_DS_2_-VASc score. The group of patients with high risk of thromboembolic complications comprises men who receive at least 2 points and women with at least 3 points in the CHA_2_DS_2_-VASc score [6]. Most patients with AF are at high risk of thromboembolic complications [7,8]. According to the current guidelines of the European Society of Cardiology (ESC), AF patients with a high risk of thromboembolic complications should receive anticoagulant treatment [9]. However, not all patients receive this treatment. 

The aim of this study was to evaluate the frequency of oral anticoagulant (OAC) use in AF patients with a high risk of thromboembolic complications and to identify factors predisposing against the use of OACs in the researched group of patients.

## 2. Methods and Materials

### 2.1. Study Population

The presented study was written on the basis of the Polish, multicenter, prospective Polish Atrial Fibrillation Registry (POL-AF) comprising 10 cardiology hospitals (ClinicalTrials.gov: NCT04419012). The recruitment period lasted from 1 January 2019 to 1 December 2019, and subsequently hospitalized patients diagnosed with AF joined the study. The study’s inclusion criteria were: diagnosed AF and age ≥ 18 years. Patients who died during hospitalization and those with valvular AF (valve prosthesis, at least moderate mitral stenosis) were excluded from the study. Additionally, patients hospitalized in order to have AF substrate ablation were not included. The data concerning comorbidities, laboratory research results, and anticoagulant treatments were evaluated.

Patients with a high risk of thromboembolic complications were included in the presented study. In total, after adopting the aforementioned exclusion criteria, 3614 patients were included (Figure 1).

### 2.2. Covariates

Researchers collected data regarding medical records, demographics, diagnostic test results, AF type, and pharmacotherapy.

The HAS-BLED (Hypertension, Abnormal Renal/Liver Function, Stroke, Bleeding, Labile INR, Elderly (>65 years), Drug/Alcohol Consumption) score was used to assess bleeding risk [10].

The CKD-EPI equation was applied to calculate the estimated glomerular filtration rate (eGFR), which was used to evaluate the function of patients’ kidneys.

The Ethics Committee of the Świętokrzyska Medical Chamber in Kielce (104/2018) approved the study. It also waived the requirement of obtaining informed consent from the patients.

### 2.3. Stroke Risk Assessment

Thromboembolic risk was defined according to the CHA_2_DS_2_-VASc (Congestive Heart Failure, Hypertension, Age ≥ 75 years, Diabetes Mellitus, Stroke/Transient Ischemic Attack, Vascular Disease, Age 65–74 Years, Sex Category) score [11,12,13,14].

Stroke risk was assessed using the CHA_2_DS_2_-VASc score and was categorized as low (score 0 for males, 1 for females), intermediate (score 1 for males and 2 for females) and high risk (score ≥ 2 for males and ≥3 for females). Patients with a high thromboembolic complication risk according to their CHA_2_DS_2_-VASc score were included in the study.

### 2.4. Stroke Prevention Assessment

An evaluation of antithrombotic therapy advised during patients’ hospital discharge was made. It was possible to define the following four types of regimen: no antithrombotic treatment, OAC ± antiplatelet drug (APT), APT only, and heparin. The OAC group included vitamin K antagonist (VKA), apixaban, rivaroxaban, and dabigatran. Despite being registered in Europe as a pharmaceutical against thromboembolic complications in AF patients, edoxaban was not obtainable in Poland. The APT group included ticagrelor, acetylsalicylic acid and/or clopidogrel, and prasugrel.

### 2.5. Statistical Analyses

Statistical analysis was conducted with the use of the STATISTICA 13.3 statistical package. Quantitative variables were presented as mean values with the standard deviations, whereas qualitative data were presented as numbers and percentages.

Next, multivariate logistic regression was performed to calculate multivariate odds ratios together with 95% confidence intervals. For that purpose, continuous variables such as age, hemoglobin level, and estimated glomerular filtration rate were changed into categorical variables. In the analysis, *p* < 0.05 was considered statistically significant.

## 3. Results

### 3.1. Baseline Characteristics

Among the 3614 patients included in the analysis, the mean age of the total population was 73.6 ± 10.3 years, where 43.5% were female. Hypertension was the most common comorbidity (87.4%), and 70% of the patients had a heart failure. Among non-cardiac comorbidities, impaired renal function was the most common diagnosis (47.9%). The most commonly reported AF type was paroxysmal AF (47.7%), whereas 30.1% of patients had a permanent AF.

In the present study, all patients had a high stroke risk, and 52.8% of the patients had a CHA_2_DS_2_-VASc score ≥ 5 points. High bleeding risk according to the HAS-BLED score was diagnosed in 33.4% of the patients.

The patients from the study group were most often admitted to hospitals in need of electrical cardioversion (22%) and due to worsening of heart failure (21.8%).

Baseline characteristics of patients according to antithrombotic strategies are presented in Table 1.

### 3.2. Antithrombotic Therapy Use

In the presented study, 3306 patients (91.5%) received OACs, 135 patients (3.7%) received APT,96 (2.7%) received heparin, and 77 (2.1%) were without OACs or APT.

In the group treated with OACs, 603 patients received VKA, and 2703 (82%) were treated with non-vitamin K antagonist oral anticoagulants (NOACs). Among NOAC-treated patients, 1076 (39.8%) were administered rivaroxaban, 893 (33%) received apixaban, and 734 (27.2%) received dabigatran.

In the study group, 1059 patients (39.2%) received a reduced NOAC dose. Reduced NOAC was also applied in 321 dabigatran patients (43.7%), 409 rivaroxaban patients (38%), and 329 apixaban patients (36.8%). Appropriate NOAC dose reduction was observed in 769 patients (72.6%), and inappropriate NOAC dose reduction was observed in 242 patients (22.9%). The remaining 48 patients (4.5%) lacked data allowing the assessment of the appropriateness of the reduced NOAC dose choice. Figure 2 shows the OAC prescription based on the CHA_2_DS_2_-VASc score and Figure 3 shows the prescription of OACs based on the HAS-BLED score.

### 3.3. Predictors of the Individual Stroke Prevention Use

During the analysis of individual antithrombotic strategy selections, it was possible to create logistic regression models for OACs versus no OACs.

The univariate logistic regression analysis showed multiple predictors of a specific OAC choice (Appendix A). In the multivariable model, factors linked to the prescription of an OAC included the following: age ≥ 75, hypertension, previous myocardial infarction, peripheral arterial disease, gastrointestinal bleeding, intracranial bleeding, cancer, hospitalization due to electrical cardioversion, hospitalization due to acute coronary syndromes, hemoglobin < 12 g/dL, and eGFR < 60 mL/min/1.73 m^2^.

Table 2 demonstrates predictors of the use of OAC. Independent predictors of the OAC use included hospitalization due to electrical cardioversion (OR 6.02, 95%CI 3.32–10.89, *p* < 0.001) and hypertension (OR 1.40, 95%CI 1.01–1.95, *p* = 0.049). Intracranial bleeding (OR 0.15, 95%CI 0.07–0.35, *p* < 0.001), gastrointestinal bleeding (OR 0.25, 95%CI 0.17–0.37, *p* < 0.001), cancer (OR 0.37, 95%CI 0.25–0.55, *p* < 0.001), hospitalization due to acute coronary syndromes (OR 0.48, 95%CI 0.33–0.69, *p* < 0.01), and hemoglobin < 12 g/dL (OR 0.62, 95%CI 0.48–0.81, *p* < 0.01) decreased the likelihood of using OACs.

## 4. Discussion

The present study provides significant insight into antithrombotic therapy in high-stroke-risk patients with AF. The main observations are as follows. OAC non-prescription in stroke prevention in high-risk patients with AF was low. A high percentage of patients administered anticoagulants were treated with NOACs. We identified factors associated with a decreased likelihood of OAC prescription, and all were associated with high bleeding risk.

According to the guidelines of the European Society of Cardiology (ESC) as well as expert documents, it is advisable to use OACs in AF patients with a high risk of thromboembolic complications [9,10,11,12,13]. In some percent of AF patients there are contraindications to the use of OACs, and therefore it will never happen in the real world that all patients with AF who are recommended OACs will take them. In the present study, OACs were not used in 8.5% of AF patients with high thromboembolic complication risk. A comparison of our observations to the findings of other established AF registries indicates that there are principal regional differences in the prescription of OACs, and that it varies widely depending on the study period and study population. In a Korean population of high-stroke-risk AF patients, 17% were prescribed no antithrombotic therapy [14]. In the National Cardiovascular Data Registry (NCDR)’s Practice Innovation and Clinical Excellence (PINNACLE) Registry involving 674,841 AF patients of high stroke risk, authors noted that 43% of patients did not receive OACs, although the proportion of those without OAC treatment varied considerably within clinically relevant strata [15]. Among AF patients with CHA_2_DS_2_-VASc ≥ 2, 31% and 13% of patients in the Global Anticoagulant Registry in the FIELD-Atrial Fibrillation (GARFIED-AF) and Outcomes Registry for Better Informed Treatment of Atrial Fibrillation (ORBIT-AF) II, respectively, were not treated with OACs. Among these patients, there was significant geographic variability in the non-use of OACs across countries, from 69% to 7% in GARFIELD-AF; and across states within the United States, from 34% to 0% in ORBIT-AF II [16]. The differences between European, American, and global registries in the use of OACs can to some extent be connected with the differences in the researched populations. Europe’s higher OAC use could result from more frequent NOAC use. This, in turn, may be linked to an explicit class I recommendation for the application of NOACs instead of VKAs included in ESC guidelines. Temporal trends in OAC prescription were observed. In the present study including Polish AF patients with a high stroke risk hospitalized in 2018, the percentage of patients treated with OACs was very high, and in another study also involving Polish AF patients but in the years 2004–2012, the percentage of patients treated with OACs was lower, at 65% [17].

This study showed promising trends in oral anticoagulation for AF according to NOAC prescription. Major and randomized clinical trials have demonstrated the non-inferiority or superiority of each NOAC compared to VKAs for stroke prevention [18,19,20]. A meta-analysis of these trials demonstrated very favorable risk–benefit profiles for NOACs versus VKAs [21]. The introduction of NOACs in 2010 changed the landscape of stroke prevention in AF. In our study, most patients with a high risk of stroke were treated with NOACs. These results were in line with the results from other studies. In GARFIELD-AF, the percentage of prescribed NOACs increased from 34% to 62% in 3 years [22]. The EORP-AF General Long-Term General Registry, in comparison to EORP-AF Pilot, indicated that over the course of four years there was a rise of NOAC prescription from less than 10% to about 35% of patients [21,23]. Observing temporal trends in anticoagulant treatment, it is possible to forecast a further increase of the percentage of patients with AF who will receive NOACs as a preventive treatment against thromboembolic complications. Therefore, NOACs, which have a better safety profile than VKAs, have become the agents of choice for patients who have not previously received antithrombotic or APT treatment.

Our observations indicate that clinicians are can identify the patients most appropriate for OAC treatment, and therefore administer this treatment to the most suitable candidates. However, among patients with high bleeding risk, the percentage of individuals for which OAC use was recommended was significantly lower. Indeed, in the present study, previous intracranial or gastrointestinal bleeding and specific risk bleeding factors such as cancer, anemia, and hospitalization due to acute coronary syndrome were connected with a lack of OAC prescription. History of intracranial bleeding was the strongest predictor of OAC non-prescription. Similarly, Lee et al. [8] showed that a history of intracranial hemorrhage was associated with OAC underuse. The decision to include OACs after past intracranial bleeding is not easy. The pivotal clinical trials of all four NOACs excluded patients with a prior history of intracranial bleeding. In the analysis of patients after intracranial bleeding who started to receive OACs, it was shown that NOACs were associated with a significantly lower risk of intracranial bleeding compared to warfarin [24]. In the present study, gastrointestinal tract bleeding was also an important factor connected to OAC underuse. As in the study of Hess et al. [25], OACs were recommended less frequently in the group of patients with a high risk of thromboembolic complications who had a prior thromboembolic complication. Another factor connected with the restriction of OAC in patients indicated to obtain such a treatment was hospitalization due to acute coronary syndrome, and the associated necessity to use antiplatelet treatment. It appears that in clinical practice, adjusting the doses of NOACs or VKAs based on a clinical risk–benefit balance can result in problems when administering OACs and antiplatelet pharmaceuticals simultaneously. It is essential to observe that although the risk factors of bleeding are often included in composite bleeding risk scores such as the HAS-BLED score, current guidelines do not suggest withholding OACs due to a high predicted bleeding risk. Altogether, our findings underline persistent concern about bleeding complications in patients treated with OACs and emphasize the necessity for a better understanding of optimal stroke prevention strategies in AF patients with high stroke and bleeding risks.

## 5. Study Strengths and Limitations

The present study includes a unique description of clinical data from Polish AF populations rather than data from selected or registered patients from trials. Our findings show the real-world clinical practice pattern of antithrombotic strategy in AF patients. Several limitations of our study must be emphasized, however. Firstly, due to the lack of long-term patient observation, it was not possible to assess long-term prognosis in AF patients treated with an individual antithrombotic strategy. Secondly, there were hospitalized AF patients examined where only some of them had a first-time diagnosed AF, and only some of them started an anticoagulant treatment. Therefore, despite the registry referring to hospitalized patients, the anticoagulant therapy in most of them began in ambulatory conditions prior to hospital admission. Patients hospitalized to undergo AF ablation were not included in the registry for two reasons: first of all, catheter ablation is not performed in all centers; secondly, it was recognized that a clinical profile of patients undergoing ablation due to AF is different from most AF patients in that they are younger and do not have concomitant diseases.

## 6. Conclusions

In the largest dedicated registry of Polish hospitalized patients with AF, the majority of patients with high stroke risk were treated with OACs. Factors associated with the absence of OACs were correlated with elevated bleeding risk, and previous bleeding was among the most important factors.

## Figures and Tables

**Figure 1 jcm-10-04611-f001:**
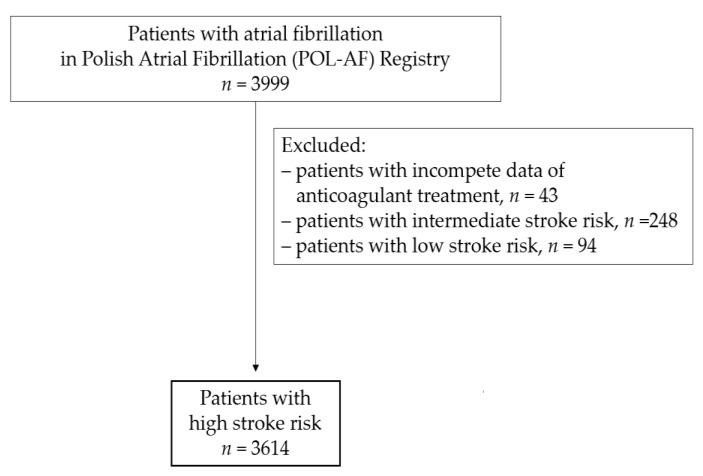
Study flow chart.

**Figure 2 jcm-10-04611-f002:**
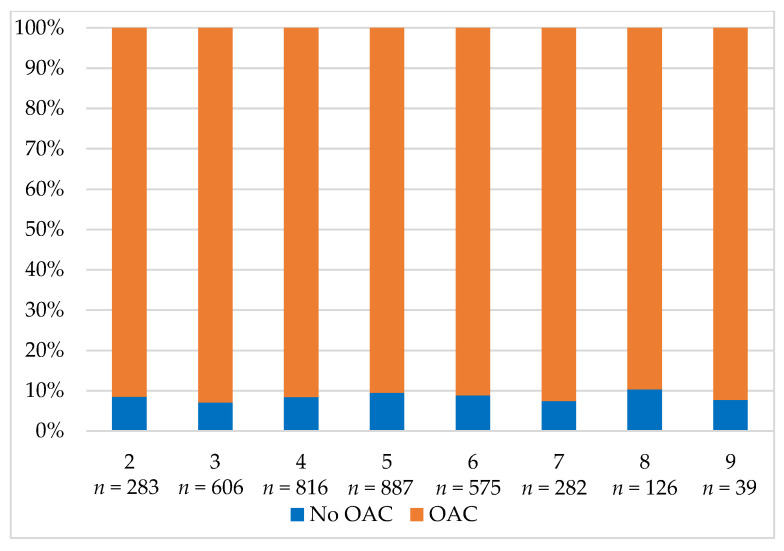
The prescription of OAC based on the CHA_2_DS_2_-VASc score. Abbreviation: OAC, oral anticoagulant.

**Figure 3 jcm-10-04611-f003:**
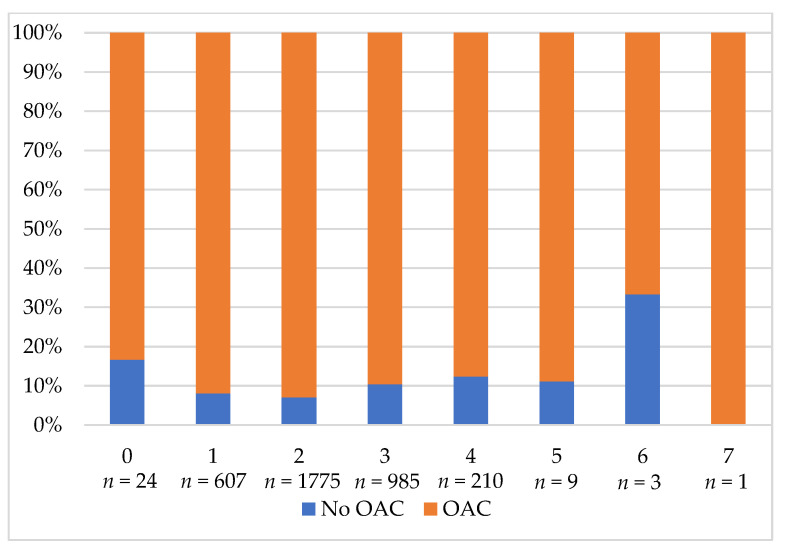
The prescription of OAC based on the HAS-BLED score. Abbreviation: OAC, oral anticoagulant.

**Table 1 jcm-10-04611-t001:** Baseline characteristics of the study group according to antithrombotic strategies.

Clinical Characteristic	All *n* = 3614	OAC *n* = 3306	No OAC *n* = 308		
			APT *n* = 135	Heparin *n* = 96	Without any Stroke Prophylaxis *n* = 77
Age mean (SD), years	73.6 (10.3)	73.5 (10.2)	74.6 (10.3)	72.6 (10.2)	77.1 (12.1)
<65	598 (16.5)	555 (16.8)	18 (13.3)	15 (15.9)	10 (13.3)
65–74	1285 (35.6)	1188 (35.9)	46 (34.1)	39 (40.6)	24 (31.2)
≥75	1731 (47.9)	1563 (47.3)	71 (52.6)	42 (43.8)	43 (55.8)
Female	1572 (43.5)	1445 (43.7)	60 (44.4)	48 (50.0)	32 (41.6)
Type of atrial fibrillation					
Paroxysmal	1723 (47.7)	1569 (47.5)	80 (59.3)	57 (59.4)	35 (45.5)
Persistent	803 (22.2)	760 (23.0)	14 (10.4)	15 (15.6)	15 (19.5)
Permanent	1088 (30.1)	977 (29.6)	41 (30.4)	24 (25.0)	27 (35.1)
Medical history					
Hypertension	3174 (87.4)	2918 (88.3)	112 (83.0)	83 (86.5)	63 (81.8)
Heart failure	2529 (70.0)	2311 (69.9)	92 (68.1)	60 (62.5)	50 (64.9)
Vascular disease	2207 (61.1)	2005 (60.6)	112 (83.0)	64 (66.7)	42 (54.5)
Coronary artery disease	1979 (54.8)	1796 (54.3)	107 (79.3)	59 (61.5)	36 (46.8)
Previous myocardial infarction	881 (24.4)	782 (23.7)	60 (44.4)	25 (26.0)	18 (23.4)
Peripheral artery disease	564 (15.6)	500 (15.1)	32 (23.7)	21 (21.9)	11 (14.3)
Previous stroke/TIA/systemic embolism	648 (17.9)	599 (18.1)	25 (18.5)	28 (29.2)	8 (10.4)
Diabetes mellitus	1341 (37.1)	1218 (36.8)	53 (39.3)	35 (36.5)	32 (41.6)
Any previous bleeding	118 (3.3)	93 (2.8)	14 (10.4)	3 (3.1)	4 (5.2)
Intracranial bleeding	29 (0.8)	18 (0.5)	4 (3.0)	0 (0.0)	2 (2.6)
Gastrointestinal bleeding	149 (4.1)	138 (4.2)	4 (3.0)	2 (2.1)	4 (5.2)
Cancer	186 (5.1)	149 (4.5)	13 (9.6)	8 (8.3)	5 (6.5)
Hemoglobin < 12 g/dL	872 (24.1)	751 (22.7)	45 (33.3)	25 (26.0)	30 (39.0)
eGFR < 60 mL/min/1.73 m^2^	1731 (47.9)	1555 (47.0)	80 (59.3)	40 (41.7)	40 (51.9)
eGFR < 30 mL/min/1.73 m^2^	255 (6.2)	178 (5.4)	24 (17.8)	11 (11.5)	12 (15.6)
Thromboembolic risk					
CHA_2_DS_2_VASc score mean (SD)	4.7 (1.6)	4.7 (1.6)	4.9 (1.6)	4.9 (1.6)	4.5 (1.4)
≥3	3331 (92.2)	3047 (92.2)	128 (94.8)	90 (92.8)	72 (93.5)
≥5	1909 (52.8)	1737 (52.5)	80 (59.3)	57 (59.4)	39 (50.6)
Bleeding risk					
HAS-BLED score mean (SD)	2.2 (0.8)	2.2 (0.8)	2.3 (0.9)	2.21 (0.9)	2.2 (0.8)
≥3	1208 (33.4)	1078 (32.6)	58 (43.0)	33 (34.4)	27 (35.1)
≥5	13 (0.4)	11 (0.3)	1 (0.7)	0 (0.0)	0 (0.0)
Reason for hospitalization					
Electrical cardioversion	796 (22.0)	784 (23.7)	5 (3.7)	21 (21.9)	4 (5.2)
Planned coronarography/PCI	372 (10.3)	338 (10.2)	21 (15.6)	7 (7.3)	5 (6.5)
CIED implantation/reimplantation	346 (9.6)	329 (10.0)	5 (3.7)	11 (11.5)	5 (6.5)
Acute coronary syndrome	240 (6.6)	197 (6.0)	38 (28.1)	7 (7.3)	4 (5.2)
Heart failure	788 (21.8)	714 (21.6)	24 (17.8)	16 (16.7)	23 (29.9)
Ablation other than AF	189 (5.2)	172 (5.2)	8 (5.9)	7 (7.3)	7 (9.1)
AF without any procedures	191 (5.3)	180 (5.4)	4 (3.0)	5 (5.2)	3 (3.9)

Abbreviations: AF, atrial fibrillation; APT, antiplatelet drug; CIED, cardiac implantable electronic device; eGFR, estimated glomerular filtration rate; IQR, interquartile range; OAC, oral anticoagulant; PCI, percutaneous coronary intervention; SD, standard deviation; TIA, transient ischemic attack.

**Table 2 jcm-10-04611-t002:** Factors associated with the selection of an OAC over no OAC for stroke prevention in AF patients: multivariable logistic regression models.

Factors	OAC Versus No OAC		
	OR	95%CI	*p*
Hospitalization due to electrical cardioversion	6.02	3.32–10.89	<0.001
Hypertension	1.40	1.01–1.95	0.049
Age ≥ 75	1.06	0.82–1.36	0.701
Myocardial infarction	0.89	0.68–1.17	0.400
Peripheral artery disease	0.88	0.64–1.21	0.411
Intracranial bleeding	0.15	0.07–0.35	<0.001
Gastrointestinal bleeding	0.25	0.17–0.37	<0.001
Cancer	0.37	0.25–0.55	<0.001
Hospitalization due to acute coronary syndrome	0.48	0.33–0.69	<0.001
Hemoglobin < 12 g/dL	0.62	0.48–0.81	<0.001
eGFR < 60 mL/min/1.73 m^2^	0.86	0.67–1.11	0.238

Abbreviation: CI, confidence interval; eGFR, estimated glomerular filtration rate; OAC, oral anticoagulant; OR, odds ratio.

## Data Availability

Data are contained within the article.

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
