# Peer review of "Why Did All Patients with Atrial Fibrillation and High Risk of Stroke Not Receive Oral Anticoagulants? Results of the Polish Atrial Fibrillation (POL-AF) Registry"

_jcm, 2021, doi:10.3390/jcm10194611_

Round 1

Reviewer 1 Report

Excellent study for better understanding risks of no OAC treatment in high risk groups of patients.

  1. I would like to note that in Table 1 «No OAC» should logically include APT, Heparin and no antithrombotic patients. It would be more comprehensive if the authors make No OAC (n=308) common block and divide it into three parts: APT (n=135), Heparin (n=96) and no treatment (n=77). Otherwise, it’s not very clear.
  1. Line 137: it’s better to say not «...because of heart failure» but «...due to worsening of heart failure.
  1. In table 1: it’s better to use the term «systemic embolism» instead of «peripheral embolism».

Author Response

Response to Reviewer 1

Dear Reviewer,

I am pleased to resubmit for publication the revised version of Why Did not Patients with Atrial Fibrillation and High Risk of Stroke Receive Oral Anticoagulants? Data from the POLish Atrial Fibrillation (POL-AF) Registry.

The reviewer’s comments were very helpful and greatly appreciated. We have addressed each concern and hope that this revised manuscript is now acceptable. Each comment is discussed in detail below. Revisions are indicated using the “Track  Changes” function. Thank you for allowing us to resubmit our manuscript.

Point 1: I would like to note that in Table 1 «No OAC» should logically include APT, Heparin and no antithrombotic patients. It would be more comprehensive if the authors make No OAC (n=308) common block and divide it into three parts: APT (n=135), Heparin (n=96) and no treatment (n=77). Otherwise, it’s not very clear.

Response 1: Table 1 has been changed according to mentioned above instructions.

Point 2: Line 137: it’s better to say not «...because of heart failure» but «...due to worsening of heart failure.

Response 2: The term «due to worsening of heart failure» was used instead of «because of heart failure».

Point 3: In table 1: it’s better to use the term «systemic embolism» instead of «peripheral embolism»

Response 3: The term «systemic embolism» was used instead of «peripheral embolism».

Thank you for the review and guidelines and we hope that now you will find our revised manuscript suitable for publication.

Kind Regards

Authors

Reviewer 2 Report

Anna Szpotowicz and colleagues present an interesting and clinically important article titled “Why did patients with atrial fibrillation and high risk of stroke not receive oral anticoagulants? Results of the POLish atrial fibrillation (POL-AF)”.

While this article is well and carefully described and results and conclusions were understandable, there are several aspects to be published in the Journal.

To answer the questions of the title name of this article, study population is relatively restricted subjects. Study population is the patients admitted to cardiology hospitals for some reasons. This is not the general, and is limited populations.

My questions are as follows;

  1. In the tile, the authors ask why the patients with AF and high risk of stroke not receive OACs. However, in the results, most patients (91.5% of include patients) use OACs. I think this high rate of DOACs utilization is contradictory to the “title name”. Therefore, title name of this article should be a more appropriate one.
  2. In high age patients with AF, severely decreased renal function (eGFR < 30 ml/min/1.73 cm2) is one of the important reasons to select OACs (DOAC, warfarin or no-OAC). If possible, add this category in the Table 1.
  3. Reduced DOAC dose was used in the case of 1,059 patients (39.2%). Do these low dose prescriptions meet reduced dose criteria or off-label use? And, are there any reasons to use low dose DOAC?

Author Response

Response to Reviewer 2  

Dear Reviewer,

I am pleased to resubmit for publication the revised version of Why Did not Patients with Atrial Fibrillation and High Risk of Stroke Receive Oral Anticoagulants? Data from the POLish Atrial Fibrillation (POL-AF) Registry.

The reviewer’s comments were very helpful and greatly appreciated. We have addressed each concern and hope that this revised manuscript is now acceptable. Each comment is discussed in detail below. Revisions are indicated using the “Track  Changes” function. Thank you for allowing us to resubmit our manuscript.

Point 1: In the tile, the authors ask why the patients with AF and high risk of stroke not receive OACs. However, in the results, most patients (91.5% of include patients) use OACs. I think this high rate of DOACs utilization is contradictory to the “title name”. Therefore, title name of this article should be a more appropriate one.

Response 1: The title of the article has been changed.

Point 2: In high age patients with AF, severely decreased renal function (eGFR < 30 ml/min/1.73 cm2) is one of the important reasons to select OACs (DOAC, warfarin or no-OAC). If possible, add this category in the Table 1.

Response 2: Category eGFR<30 ml/min/1.73 mwas added in table 1.

Point 3: Reduced DOAC dose was used in the case of 1,059 patients (39.2%). Do these low dose prescriptions meet reduced dose criteria or off-label use? And, are there any reasons to use low dose DOAC?

Response 3: Data concerning usage of reduced dose of DOAC was added in section Results. The manuscript on this issue entitled „Inappropriate prescription of the reduced dose of NOAC in clinical practice -  results of the POLish Atrial Fibrillation Registry (POL-AF) in hospitalised patients” is in the reviews.

Thank you for the review and guidelines and we hope that now you will find our revised manuscript suitable for publication.

Kind Regards

Authors

Round 2

Reviewer 2 Report

The authors improved the sentences. There is nothing more to point out.

This manuscript is a resubmission of an earlier submission. The following is a list of the peer review reports and author responses from that submission.